# Affinity enrichment of extracellular vesicles from plasma reveals mRNA changes associated with acute ischemic stroke

Harshani Wijerathne[1,2], Malgorzata A. Witek [1,2,3,4], Joshua M. Jackson[1,2], Virginia Brown[2,5], Mateusz L. Hupert[6], Kristina Herrera[4], Cameron Kramer[1,2], Abigail E. Davidow[1,2], Yan Li[7], Alison E. Baird[7], Michael C. Murphy[2,8] & Steven A. Soper [1,2,3,5,6,9✉]

Currently there is no in vitro diagnostic test for acute ischemic stroke (AIS), yet rapid diagnosis is crucial for effective thrombolytic treatment. We previously demonstrated the utility of CD8(+) T-cells' mRNA expression for AIS detection; however extracellular vesicles (EVs) were not evaluated as a source of mRNA for AIS testing. We now report a microfluidic device for the rapid and efficient affinity-enrichment of CD8(+) EVs and subsequent EV's mRNA analysis using droplet digital PCR (ddPCR). The microfluidic device contains a dense array of micropillars modified with anti-CD8α monoclonal antibodies that enriched 158 ± 10 nm sized EVs at 4.3 ± 2.1 × $10^9$ particles/100 μL of plasma. Analysis of mRNA from CD8(+) EVs and their parental T-cells revealed correlation in the expression for AIS-specific genes in both cell lines and healthy donors. In a blinded study, 80% test positivity for AIS patients and controls was revealed with a total analysis time of 3.7 h.

[1] Department of Chemistry, The University of Kansas, Lawrence, KS 66047, USA. [2] Center of BioModular Multiscale Systems for Precision Medicine, The University of Kansas, Lawrence, KS 66047, USA. [3] The University of Kansas Medical Center, Cancer Center, Kansas City, KS 66160, USA. [4] Department of Chemistry, The University of North Carolina, Chapel Hill, NC 27599, USA. [5] Bioengineering Program, The University of Kansas, Lawrence, KS 66047, USA. [6] Biofluidica, Inc., San Diego, CA 92121, USA. [7] SUNY Downstate Medical Center, Brooklyn, NY 11203, USA. [8] Department of Mechanical Engineering, Louisiana State University, Baton Rouge, LA 70805, USA. [9] Department of Mechanical Engineering, The University of Kansas, Lawrence, KS 66047, USA. ✉email: ssoper@ku.edu

Stroke is the leading cause of long-term disability in the United States and the fourth leading cause of death in 2018 with ~800,000 people experiencing a new or recurrent stroke each year[1]. Worldwide, stroke is responsible for ~12% of fatalities, which makes it the second leading global cause of death behind heart disease. Although the overall risk of stroke has dropped ~25% during the last decade[2], disability caused by strokes have become one of the major health problems worldwide.

Among the two types of stroke, hemorrhagic (bleeding into the brain) and acute ischemic stroke (AIS; blockage of a blood vessel), 85% of the patients experience AIS, and rapid diagnosis is essential. Clot-busting thrombolytic treatment using recombinant tissue plasminogen activator (rt-PA) is the cornerstone of AIS therapy, but is absolutely contraindicated for hemorrhagic stroke[3]. Unfortunately, rt-PA, which was approved by the US FDA in 1996, reaches only ~5% of AIS patients largely because the timeframe for treatment is 4.5 h after the onset of stroke symptoms and current tests for AIS cannot in most cases meet that time constraint[4].

The current standard-of-care for stroke patients is to undergo computed tomography (CT) to rule out hemorrhagic stroke. CT clearly shows hyper-dense lesions due to hemorrhagic stroke but is much less sensitive for AIS and other medical conditions, such as stroke mimics and prior infarctions. As such, CT has only 26% clinical sensitivity for AIS[5]. Magnetic resonance imaging (MRI) is more sensitive (83%) for both AIS and hemorrhagic stroke, but is more time consuming than CT and is not widely used in an emergency setting. In addition, many hospitals do not have the required instrumentation and trained personnel for MRI to provide real-time scanning and interpretation.

Unfortunately, there is no in vitro diagnostic for AIS in spite of the potential benefits of such a test, especially a peripheral blood-based one (i.e., liquid biopsy). Certain leukocyte subpopulations respond to AIS via dysregulation of genes[6,7]. CD8(+) T-cells were, for example, shown to contribute to inflammatory responses with AIS-associated mRNA expression observed <3 h following stroke onset[6]. Importantly, because mRNA transcription precedes protein translation, mRNA expression changes can be observed 6–12 h earlier than detection of proteins in the peripheral blood that may be responding to AIS[8,9]. Hence, assays based on CD8(+) leukocyte mRNA expression take advantage of the reduced latency of these AIS-specific markers appearing in peripheral blood[7].

Recently, extracellular vesicles (EVs) have garnered attention as biomarkers for disease diagnostics. EVs are a collection of cell-derived membranous structures comprising exosomes, ectosomes, microvesicles, etc., all of which have different biogenesis and thus can contain differences in their molecular cargo. The molecular cargo of EVs (mRNA, miRNA, proteins, etc.) can represent the molecular composition of cells from which they originate[10,11]. In the central nervous system (CNS), EVs maintain normal neuronal function but are also involved in neurodegenerative diseases. For example, EVs released from CD8(+) T-cells play key roles in CNS homeostasis, stroke pathology, and subsequent stroke recovery[12]. Therefore, EVs isolated from peripheral blood could provide information on changes in the brain, including mRNA expression changes responding to AIS. Recent studies have shown that the miRNA-17 family members sourced from EVs were highly expressed in ischemic stroke patients compared to those with stroke mimics, but the cause of higher expression was assigned to a chronic sequela of cerebrovascular small vessel disease rather than ischemic stroke[13].

The contribution of peripheral blood mononuclear cells' (PBMCs') mRNA expression changes in both pre-existing and an acute response to stroke has been studied by Moore et al. indicating a partial dependence of expression changes on pre-existing vascular risk conditions[14]. Hence, there may be a contribution of both risk and response to acute stroke on the mRNA expression profile. However, this would not diminish the value of the information concerning the acute event.

We hypothesized that EVs released by CD8(+) T-cells contain gene profiles similar to their host cells after an inflammatory response, and thus could be used to potentially detect AIS. We isolated EVs derived from activated CD8(+) T-cells via affinity enrichment using a microfluidic device, named an EV micro-affinity purification (EV-MAP) device. While EVs are commonly isolated using differential centrifugation[15] with a recovery as low as 10%[16], it requires 5–12 h to complete. Precipitation techniques with polyethylene glycol (PEG) decrease processing time[17]. However, all EVs emanating from different biological cells are isolated and not just the disease-associated ones, which can mask mRNA expression differences associated with AIS onset[18].

Microfluidics can provide efficient isolation of EVs[17]. Indeed, microfluidic devices that affinity select EVs either using microbeads packed into microchannels[19] or modifying surfaces to covalently tether capture antibodies to the microfluidic have been demonstrated[18]. In comparison with these reports, our EV-MAP possesses the throughput capacity and EV load to meet the time constraints associated with AIS diagnostics. For example, a microfluidic chip possessing herringbone mixers[20] operated at a volumetric flow rate of 0.5 μL/min, which would require 3.3 h to process 100 μL of plasma to isolate EVs, which is near the rt-PA therapeutic time window, but does not include the time for molecular analysis[20].

Our EV-MAP possesses a high-density array of micropillars decorated with anti-CD8α monoclonal antibodies (mAbs) to target CD8(+) EVs and operating at 10 μL/min can process 100 μL of plasma in 10 min with an EV recovery of ~85%. EVs were released from the affinity surface for particle counting and/or Transmission Electron Microscopy (TEM) analysis. For mRNA profiling, EVs were lysed, total RNA (TRNA) was isolated and quantified using gel electrophoresis, cDNA synthesized, and droplet digital PCR (ddPCR) performed to obtain mRNA copy numbers from the enriched EVs. mRNA profiling was motivated by our previous work recognizing panels of specific mRNA markers that were differentially expressed in CD8(+) T-cells following AIS[7]. We performed a single-blinded proof-of-concept study of clinical plasma samples collected from both AIS patients and controls, and demonstrated 80% test positivity for the identification of AIS patients using EVs' mRNA cargo.

## Results

### EV microfluidic affinity purification.
Two types of devices containing microfluidic beds populated with pillars were used: (i) A 3-bed chip (Fig. 1a–c); and (ii) a device with a z-type configuration that addressed 7 beds in parallel (Fig. 1d, e). Both devices were made from a thermoplastic to allow for high-rate production at low cost to accommodate diagnostic applications.

The mold master for the 3-bed EV-MAP was fabricated in brass using high-precision micromilling, and devices were replicated in cyclic olefin copolymer (COC) via hot embossing[21]. The device contained ~15,000 circular pillars (100 μm diameter spaced ~15 μm apart, surface area of 6.8 cm²) with a theoretical EV load capacity of $3.5 \times 10^{10}$ particles calculated based on the surface area of the device and a monolayer hexagonal packing of EVs with a diameter of 150 nm (Table 1). With its smaller surface area, the amount of mAb required for surface loading was reduced compared to the z-chip resulting in lower assay cost and making it attractive for initial assay characterization. This chip was made from COC.

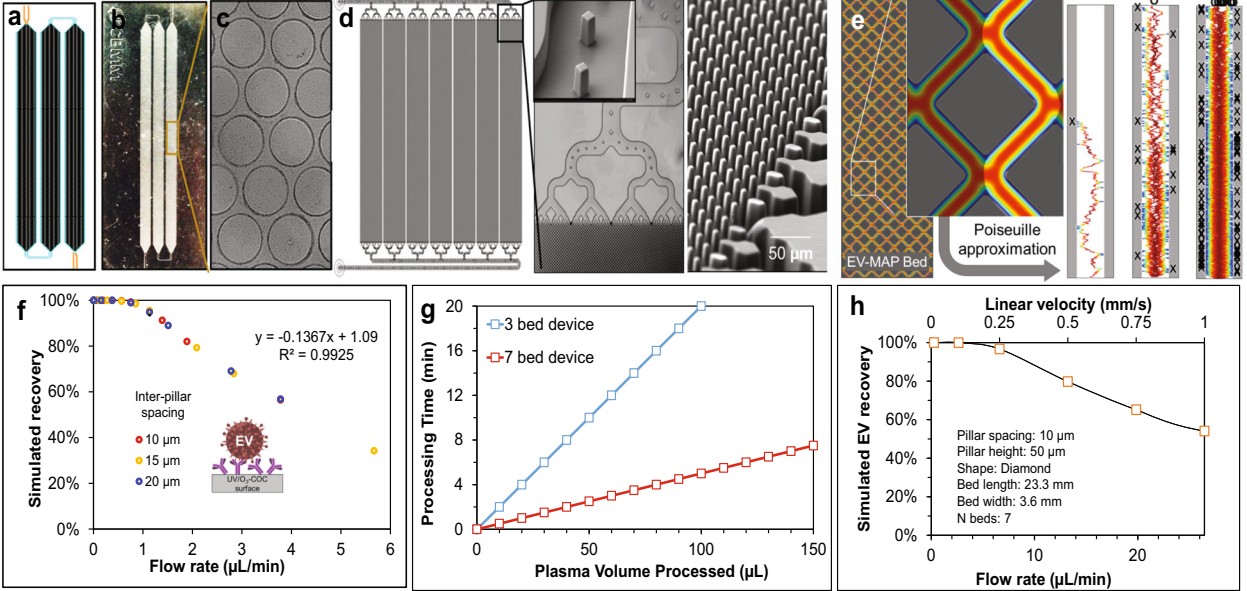

**Fig. 1 Microfluidic devices design and EV recovery. a** Picture of CAD showing the 3-bed EV-MAP with circular micropillars. **b** Hot embossed device fabricated in COC thermoplastic. **c** Circular micropillars of the device surface. **d** Picture of 7-bed EV-MAP showing the distribution channels and the diamond-shaped micropillars of the device surface. **e** Network of 10-μm microchannels between micropillars to enable efficient EV recovery by reducing the distances required for EVs to diffuse and interact with the surface-confined mAbs coated on the micropillars of the 7-bed device. The EV transport dynamics were simulated via a custom Monte Carlo model that incorporated diffusive and convective EV transfer and mAb–EV binding dynamics. Shown are tracks of individual EVs (not to scale) diffusing through a microchannel, where color scales with the EV velocity (blue-low, red-high) and "X" indicating a successful EV–mAb binding event whereas "O" indicates the EV was not captured. Results are averaged until the predicted EV recovery converges. **f** Monte Carlo simulation results for the 3-bed EV-MAP. **g** Calculated sample processing time for 3-bed (5 μL/min) and 7-bed devices (20 μL/min). **h** Results of Monte Carlo simulation for the 7-bed EV-MAP.

**Table 1 Comparison of device parameters of 3-bed and 7-bed EV-MAP.**

| Metric | 3-bed EV-MAP | 7-bed EV-MAP |
|---|---|---|
| Bed dimensions (l × w × d) (mm × mm × μm) | 122 × 1.7 × 90 | 23 × 3.6 × 50 |
| Number of pillars | 15,202 | 1,475,712 |
| Pillar geometry and dimensions (μm) | Circular 100 | Square 10 × 10 |
| Inter-pillar spacing (μm) | 15 | 10 |
| Internal surface area (cm$^2$) | 6.8 | 38.5 |
| Internal volume (μL) | 6.5 | 22.4 |
| Mass of antibody (μg) immobilized on the surface | 3.9 ± 1.3 | 23.1 ± 3.4 |
| Antibody coverage (pmole/surface area) | 26.7 ± 8.7 (~3.9 pmole/cm$^2$) | 154.0 ± 22.7 (~4.0 pmole/cm$^2$) |
| Bed capacity for EV, $d = 150$ nm | 3.5 × 10$^{10}$ particles | 2.2 × 10$^{11}$ particles |

Each z-type device contained 7 beds connected in parallel with perpendicular inlet and outlet channels arranged in a z-configuration. The device contained ~1.5 million diamond-shaped pillars (10 μm × 10 μm, 10 μm spacing) providing a -38.6 cm$^2$ surface area with a maximum theoretical particle load of 2.2 × 10$^{11}$ (Table 1). This chip was made from cyclic olefin polymer (COP).

Measurements of polymer surface hydrophilicity/hydrophobicity via water contact angle and carboxylic acid group densities via a TBO assay following activation[22] showed no significant differences between the two thermoplastics (Supplementary Figs. 1 and 2 and Supplementary Table 1) providing similar surface densities of mAbs. Covalently attached anti-CD8α antibody was removed from the chip and its concentration was evaluated using a BCA assay, which concluded that for both COC and COP the mAb surface density was ~4 pmole/cm$^2$ (Table 2).

The architecture of the EV-MAP was designed to maximize EV recovery while providing high throughput processing to keep the processing time short. In the design, sample infused into the

device moves around solid pillars with EVs diffusing laterally for potential association with mAbs immobilized onto the micropillar surfaces. Small inter-pillar spacing and long bed lengths decreased the diffusional distances and provided sufficient residence time, respectively, for increasing EV binding probability to the mAbs. To guide the design, we developed a Monte Carlo simulation that incorporated hydrodynamic Poiseuille flow, lateral and longitudinal EV diffusion, and EV-mAb binding kinetics per the Chang-Hammer model (Supplementary Tables 2 and 3)[23]. In the model, EVs were tracked until they were either bound to the surface or lost and the results were used to determine the recovery (Fig. 1e and Supplementary Figs. 3 and 4).

The 3-bed EV-MAP has an inter-pillar spacing of ~15 μm and a bed length of 122 mm with large micropillars ($d = 100$ μm) that resulted in a recovery of 41% at 5 μL/min (Fig. 1f). The 7-bed EV-MAP pillar size and inter-pillar spacing was 10 μm and yielded a 97% EV recovery at 5 μL/min (Fig. 1f). The 7-bed EV-MAP provided high sample processing capabilities in comparison with the 3-bed device. Assuming the same recovery, the 7-bed device

**Table 2 Empirical data for the protein content evaluated following affinity isolation of EVs using the 3-bed EV-MAP.**

| Volumetric flow rate at which 100 μL of healthy plasma processed | | | | | |
|---|---|---|---|---|---|
| | 0.5 μL/min | 1.0 μL/min | 2.0 μL/min | 5.0 μL/min | 10.0 μL/min |
| Protein mass (μg) | 12.7 ± 3.4 | 24.8 ± 4.0 | 13.1 ± 4.1 | 8.5 ± 4.3 | 7.5 ± 4.06 |
| **Volume of healthy plasma processed on a 3-bed EVMAP at 10 μL /min** | | | | | |
| | 0 μL | 100 μL | 300 μL | 500 μL | 1000 μL |
| Protein mass (μg)—BSA modified chip | 3.9 | 5.6 ± 2.1 | 13.7 ± 3.1 | 13.2 ± 2.1 | 12.8 ± 2.9 |
| Protein mass (μg)—anti-CD8 Ab modified chip | 3.3 | 7.5 ± 4.1 | 22.7 ± 3.0 | 96.1 ± 6.1 | 106.2 ± 7.1 |

can operate at an 8-fold higher throughput (Fig. 1g). In the pilot study presented herein, clinical samples collected from AIS patients were processed using the 7-bed EV-MAP to ensure performing ddPCR with a sufficient dynamic range and above the limit-of-detection for mRNA quantification while, at the same time, keeping the sample processing time short. The 3-bed EV-MAP was used for initial assay characterization prior to clinical testing.

Modeling experiments were verified with empirical data evaluating EV protein concentration from enriched CD8α(+) EVs from healthy donor plasma as a function of the volumetric flow rate (Table 2). A similar trend was observed in both the COMSOL simulation and experimental data with higher recoveries (based on protein content) achieved at lower volumetric flow rates. The recovery of EVs determined using a "self-referencing" method[24] was 48% for sample processed at 5 μL/min, which agreed favorably with the simulation predicted recovery of 41% at this same volumetric flow rate. The saturation of the 3-bed EV-MAP was observed for ~500 μL of healthy donor plasma processed on chip for the affinity selection of CD8α(+) EVs. The amount of protein extracted was 96 μg (Table 2). To determine the amount of protein non-specific adsorption, we modified the surface of the EV-MAP devices with bovine serum albumin (BSA) instead of anti-CD8 mAb and processed the same volume of healthy donor plasma (Table 1). The amount of protein non-specifically bound to the bed was ~10× lower, ~9 μg for 500 μL of plasma sample. Four micrograms of protein was attributed to BSA, while the remaining 5 μg of protein mass could either be from EVs or plasma proteins.

**Affinity enrichment of EVs using EV-MAP.** Anti-CD8α mAbs were covalently attached to the surfaces of the EV-MAP with an oligonucleotide bifunctional linker (Fig. 2a)[24], which contained a uracil residue that can be cleaved with USER® (Uracil Specific Excision Reagent) hence allowing for release of EVs after affinity enrichment, similar to what we have shown for affinity-enriched circulating tumor cells[24].

We evaluated the EV-MAP performance metrics by isolating and releasing EVs originating from the MOLT-3 cell line (T-cell model) cultured in EV-depleted fetal bovine serum (FBS). MOLT-3 cells showed CD8 expression at an average of 2500 CD8α receptors/cell (Supplementary Figs. 5 and 6) in 13% of the population, similar to that seen for this same cell line in previous reports[25].

Following enrichment of EVs, we used an immunoassay with APC-labeled anti-CD8α mAbs to target surface EV-CD8α antigens. Imaging showed higher fluorescence intensity when devices were modified with anti-CD8α mAbs compared to control devices, which contained no mAbs or an isotype (Fig. 2b–d).

The mAb linker cleaved with USER® released enriched EVs from the EV-MAP as confirmed by TEM and nanoparticle tracking analysis (NTA). TEM indicated the presence of EVs (Fig. 2e–g) in the device's eluent following release with NTA

indicating an average particle size of 150 ± 23 nm and a concentration of $1.6 ± 0.7 × 10^8$ particles/100 μL media ($n = 3$), suggesting we are operating below the theoretical EV load capacity of the 3-bed chip. Incubation with USER® yielded a 96.6 ± 1.3% EV release efficiency (Fig. 2h–k).

**EV mRNA expression in an inflammation model.** By evaluating the response of MOLT-3 cells and the EVs they generate when exposed to lipopolysaccharide (LPS), we mimicked the inflammation process during an AIS event[26]. LPS is a component of the outer membrane of Gram-negative bacteria that induces cell inflammation in macrophages and T-cells by releasing cytokines[27]. LPS stimulation conditions in cell culture (Fig. 3a) were optimized to eliminate potential DNA or RNA damage by reactive oxygen species generated in response to LPS[28]. The MOLT-3 cell line was used as the model because of its expression of CD8 antigens. CD8 expressing T-cells have been found to possess mRNA expression profiles indicative of AIS[7].

MOLT-3 cell viability was evaluated over a 72-h period cultured with 0, 1, and 100 ng/mL of LPS. Cells showed 80 ± 2% viability and no obvious changes in cell morphology after 24 h of stimulation with 100 ng/mL LPS (Fig. 3a). Therefore, these conditions were applied to the cell culture. Cells were processed through an anti-CD8 mAb modified sinusoidal microfluidic device (Fig. 3b)[22]. EVs in the conditioned media were affinity-enriched with the 3-bed EV-MAP also modified with anti-CD8α mAbs (see Fig. 3b and Methods).

We selected a gene cluster identified by Adamski et al.[7], which consisted of five genes: FOS, VCAN, PLBD1, MMP9, and CA4. This gene panel evaluated in CD8(+) T-cells has shown statistically significant differences ($p = 1.42 × 10^{-5}$) in mRNA expression between AIS patients and controls[7]. We used ddPCR (see Supplementary Table 4 for primer sequences that were designed to be close to the 3′ end of the mRNA) to provide absolute quantification of cDNA from the target mRNA. cDNA copies were normalized to ng TRNA quantified by gel electrophoresis (Supplementary Table 5 and Supplementary Fig. 7 for typical ddPCR results). Among the five genes profiled, CD8(+) EVs harvested from stimulated cells showed two genes (PLBD1 and FOS) that were upregulated upon LPS stimulation (Fig. 3c). In CD8(+) MOLT-3 cells, three genes (PLBD1, FOS, and VCAN) were upregulated compared to the non-LPS stimulated cells. The mRNA copy numbers in cells and EVs isolated from the medium with LPS were on average twice as abundant as unstimulated cells (slopes of 1.7 and 1.9 for MOLT-3 cells and EVs, respectively, Fig. 3e). We observed a 1:1 transcript ratio for PLBD1, FOS, MMP9, and CA4 in CD8(+) EVs and CD8(+) MOLT-3 cells in both stimulated and unstimulated conditions (0.84–1.02 slopes, Fig. 3f). VCAN appeared to be 2.5× more abundant in CD8(+) EVs than in CD8(+) MOLT-3 cells in both stimulated and unstimulated conditions (Fig. 3f). Although we could differentiate stimulated and unstimulated cells using three genes based on these model experiments, it is important to note that in clinical studies, a five-gene panel will be used, as gene expression data for

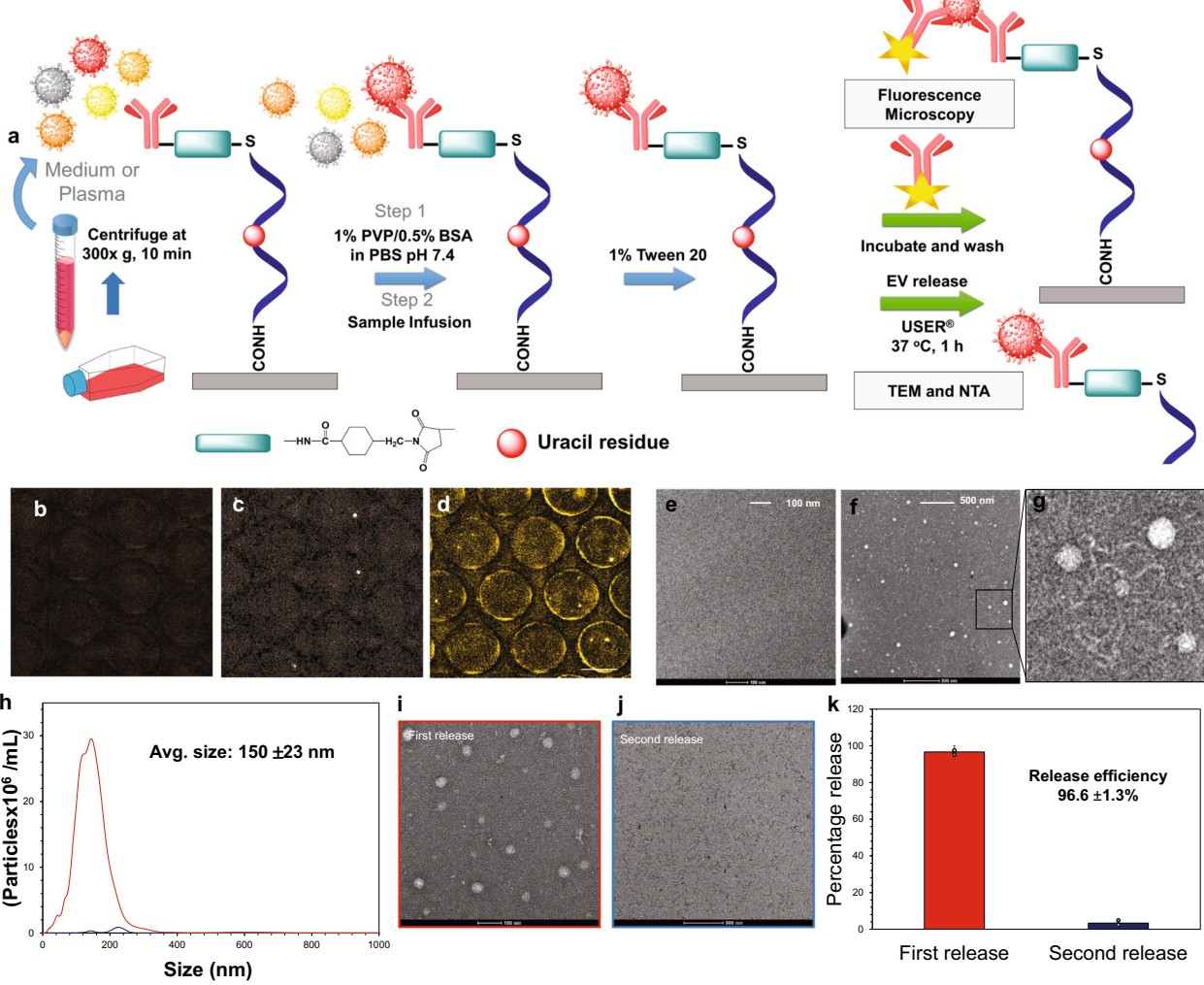

**Fig. 2 Affinity enrichment of EVs and release from the microfluidic device. a** Schematic diagram representing the workflow for sample processing and release of enriched EVs from the EV-MAP device's surface. **b** Fluorescence images after staining the EVs captured on the EV-MAP device's surface with an APC-labeled secondary antibody, Left—negative control without anti CD8α mAb; **c** isotype (IgG2B) control; **d** florescence images of CD8+ EVs captured from cell media. TEM images of: **e** USER® enzyme and buffer used for EV release from the EV-MAP device's surface with no EV infusion; **f** and **g** EVs captured and released from the MOLT-3 cell culture media. **h** NTA results (*n* = 3) and TEM images showing the number of EVs released during first (**i**) and second (**j**) USER® enzyme release. **k** Percentage of EVs released during first and second release with USER® enzyme.

leukocyte subpopulations from clinical samples have identified the five-gene panel providing the highest clinical specificity and sensitivity for AIS[7].

**Gene expression of CD8(+) T-cells and CD8(+) EVs isolated from healthy donors' plasma.** We isolated CD8(+) T-cells from healthy donor blood samples using a sinusoidal microfluidic device (Figs. 3b and 4a)[22], and enriched CD8(+) EVs from plasma from the same blood samples. Following affinity enrichment and subsequent release from the capture surface, T-cells were enumerated (~24,000 ± 3000 cells/mL of blood) and immunostained against CD45 and CD8α antigens (Supplementary Fig. 6) showing a purity of the T-cell fraction of 81.3 ± 11.5%. EVs following release were visualized and characterized using TEM (Fig. 4b).

We isolated 8.8 ng TRNA from CD8(+) T-cells enriched from 1 mL of blood as determined by TRNA gel electrophoresis. Cells' TRNA profiles were typical for eukaryotic cells (rRNA with a 28S band twice as intense as the 18S band, Fig. 4c). EVs were isolated from 500 μL of plasma using the 3-bed EV-MAP and PEG precipitation[29]. We extracted 3.3 ng and 18.1 ng EV-TRNA

following affinity isolation and PEG precipitation, respectively. Gel electrophoresis analysis of the isolated TRNA using EV-MAP and PEG indicated the presence of short RNA fragments and a lack of 28S/18S rRNA bands in the EV-enriched fractions. The RNA size ranged between 50 and 2000 nt with the highest abundance for 200 nt long fragments (Fig. 4c) indicating the presence of truncated mRNA and rRNA fragments, miRNA, and long non-coding RNAs[30,31]. We observed a positive correlation between the amount of TRNA extracted and the number of nanoparticles detected via NTA (Pearson coefficient = 0.671) and ~2 × 10$^{-18}$ g TRNA/EV particle (Fig. 4d).

Analysis of the ddPCR data for six healthy donors indicated that for all genes in our AIS panel, there was no statistical difference in expression between T-cells and their generated EVs (Fig. 4e–i). mRNA abundance for paired T-cells and their generated EVs in healthy donors show no correlation (*P* = −0.1892) unlike in the LPS-stimulated MOLT-3 cell line and its EVs (*P* = 0.8244, Fig. 3f), suggesting high similarities in mRNA transcript expression between EVs and their parental cells when inflammatory conditions were applied, as is the case with LPS-stimulated cells[32].

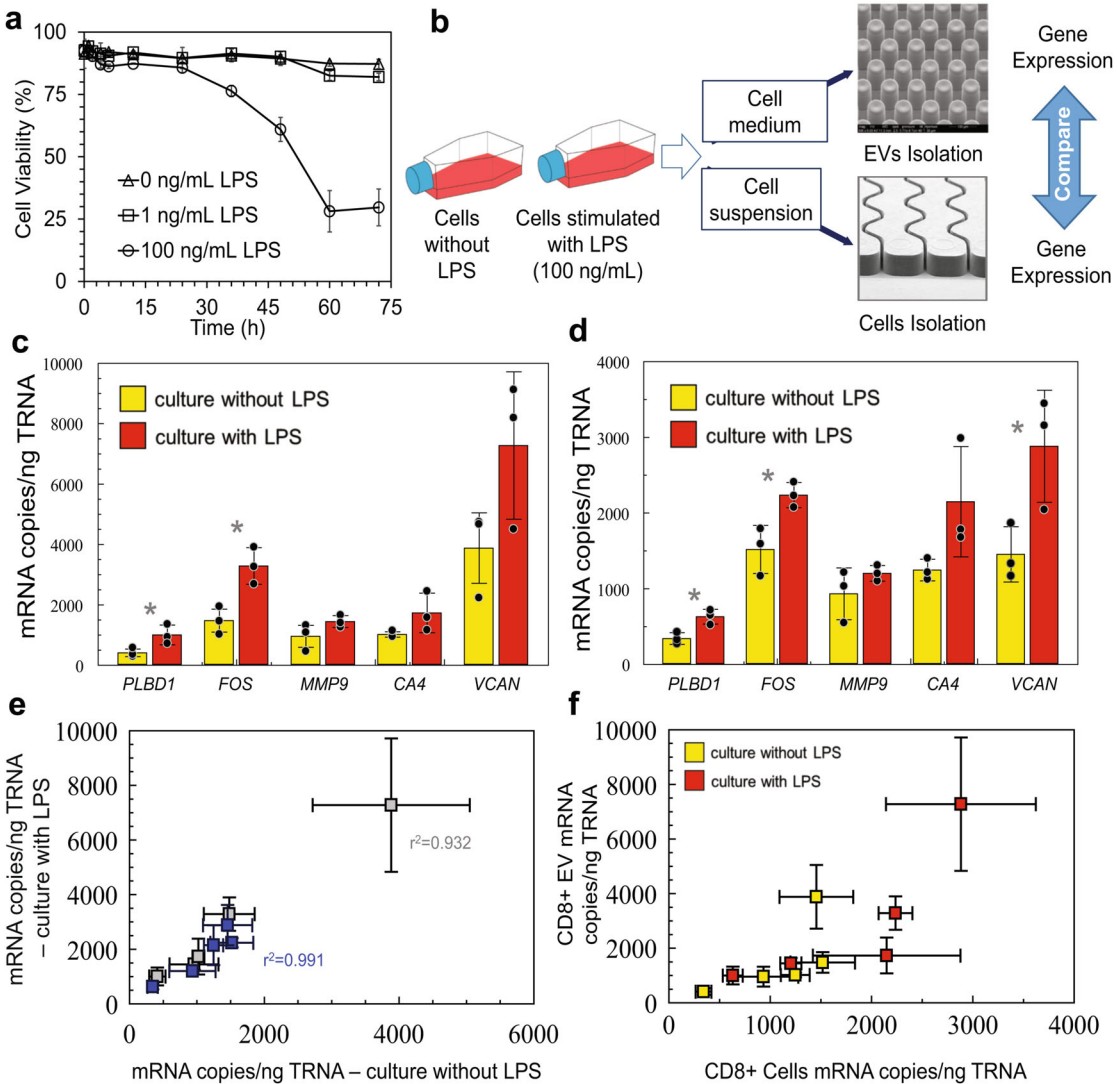

**Fig. 3 EV mRNA abundance analysis in cell line models. a** Cell line viability when cultured with different LPS concentrations in culture medium.
**b** Workflow for isolation of MOLT-3 cells and EVs from culture and gene expression analysis. mRNA gene expression profiles for CD8+ EVs cells (0.5 and
0.7 ng of RNA for stimulated and unstimulated, respectively, used in RT reactions) (**c**) and CD8+ MOLT-3 cells (0.7 ng RNA for stimulated and
unstimulated was used in RT reactions) (**d**). cDNA was diluted 5× in water before use in ddPCR. Yellow—non-stimulated conditions; red—stimulated
conditions. **e** Correlation between mRNA copies found in stimulated and unstimulated MOLT-3 cells and EVs. **f** Correlation between mRNA copies found in
EVs and MOLT-3 cells in stimulated and unstimulated conditions. (*) indicates *P* values <0.05.

**mRNA transcripts analysis in CD8(+) EVs isolated from AIS
patients and normal controls' plasma.** EVs from clinical samples
were enriched using the 7-bed EV-MAP. Patient information is
provided in Supplementary Table 6. For randomly selected
samples, we compared EV-MAP isolation to PEG EV precipita-
tion. NTA results showed a much narrower size distribution of
particles for EV-MAP (158 ± 10 nm) compared to PEG pre-
cipitation (230 ± 110 nm), which agreed with EV sizes and mor-
phology from TEM (Fig. 5b, c). ddPCR results for clinical samples
are shown in Supplementary Table 5. The yields of the TRNA
isolated from clinical samples are listed in Supplementary Table 7.
While PEG precipitation yielded more TRNA (15.1 ng) than anti-
CD8α mAb EV-MAP (4.4 ng), mRNA profiling of the isolates for
our five-gene panel differed (Fig. 5e) owing to the fact that the
PEG method cannot differentiate EVs by parental cell type ori-
gination because it isolates the entire EV population found in
plasma.

Gene expression analysis using the five-gene AIS panel was
performed for ten clinical samples to serve as a proof-of-concept

study to determine if mRNA sourced from EVs could be used to
detect AIS. The cohort of six healthy samples analyzed previously
served as a training set to differentiate between controls and AIS
samples. Gene copy numbers normalized to ng TRNA (Fig. 5g)
were analyzed by principal component analysis (PCA) and heat
map compilation for cluster determination. In PCA, sample
grouping with the training set were categorized as healthy
controls (see the box in Fig. 5i), while all others were classified as
AIS patients. Clustering was confirmed with heat maps (Fig. 5h,
i). When compared with the "key", we found 80% success in
correctly identifying patient status; samples #7 (healthy control)
and #8 (AIS patient) were misidentified. For both cohorts, there
was no statistical difference in TRNA concentration (healthy—
1.6 ng/mL average, range of 0.5–2.2 ng/mL; AIS patients—2.05
ng/mL average, range of 0.6–5.5 ng/mL). There was also no
correlation between EV particle concentration and clinical
condition (see Fig. 5f). The entire assay required 3.7 h from
sample-to-answer and is within the therapeutic time window
mandated by rt-PA treatment for AIS (4.5 h; Fig. 5j).

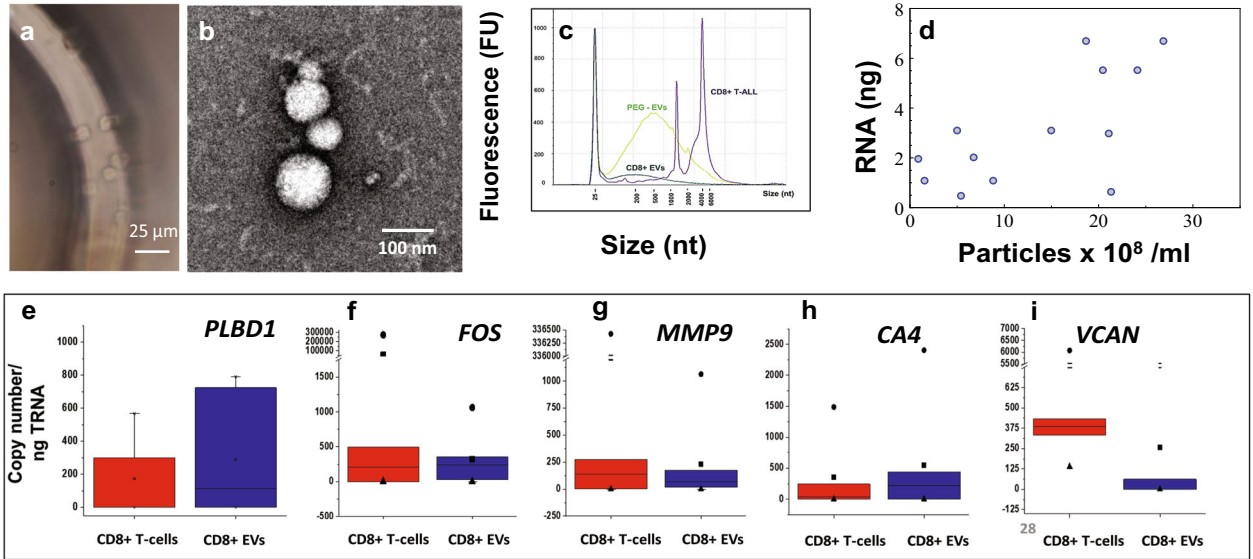

**Fig. 4 Affinity enrichment and gene expression of CD8(+) T-cells and CD8(+) EVs isolated from healthy donors. a** Micrograph of cells isolated using a curvilinear cell isolation device (bright field). **b** TEM image of EVs isolated using the EV-MAP 3-bed device. **c** Electropherogram for the separation of RNA isolated from CD8+ T-cells and EVs and PEG precipitated EVs from healthy donor plasma. **d** Correlation plot of particle concentration (presumably EVs) with RNA mass isolated from affinity-selected CD8+ EVs. Boxplots comparing the gene expression of CD8+ T-cells and CD8+ EVs isolated from healthy donor plasma for **e** PLBD1, **f** vFOS, **g** MMP9, **h** CA4, and **i** VCAN, for cells ($n = 5$) and EVs ($n = 6$); 0.7 ng and 0.8 ng of RNA isolated from cells and EV, respectively, was used in RT (+/−) reactions.

## Discussion

New blood-based biomarkers and the development of tests that utilize those biomarkers are needed to improve outcomes for AIS patients. mRNA transcripts packaged into peripheral white blood cells[7] are attractive because they can reflect the physiological perturbation imposed by AIS faster (<3 h after stroke onset) than proteins (>6 h) and thus provide indications of disease within the time constraints imposed by effective rt-PA treatment.

We have shown that expression of certain genes specific to AIS in CD8(+) T-cells provided 66%, 87%, and 100% clinical sensitivity for 2.4 h, 5 h, and 24 h following stroke onset[33], suggesting that mRNA expression differences indicative of AIS have improved clinical sensitivity with time as more leukocytes respond to an inflammatory insult. We hypothesized that gene expression changes occurring in CD8(+) T-cells responding to AIS manifest themselves into similar expression changes in CD8 (+) EVs' molecular cargo. Clinical sensitivity and specificity in assays that use blood-based biomarkers are determined by the appearance of dysregulated mRNA transcripts that allow discrimination between patients and healthy controls. The molecular processing time associated with the diagnostic is another critical factor. In its current rendition, the assay we report using EVs required 3.7 h of processing time and is within the therapeutic time window of rt-PA (Fig. 5j).

The EV-MAP platform used for enriching the EVs provided high throughput (i.e., short processing time) and sufficient EV recovery to meet the limit-of-detection requirements of ddPCR. In spite of the fact that the EV particle number is typically high, >10⁹ particles per 100 µL of sample, EV mRNA expression profiling is challenged by the fact that the mass of TRNA is low per EV particle (~2 ag per particle, Fig. 4d), and mRNA represents ~2% of the TRNA EV cargo[31]. To add to the challenges of EV mRNA expression profiling, rarely are there found full length transcripts in EVs with an exosomal origin[34]. This was evident from the fact that the TRNA found in the EV-MAP CD8α enriched fraction was predominantly <200 nt in length.

In an in vitro inflammation model that can induce potential mRNA changes during AIS, we determined high concordance in gene expression between cells and EVs (Fig. 3e) under inflammation induced with LPS (Fig. 3f)[32]. In clinical samples, $4.25 \pm 2.1 \times 10^9$ CD8 expressing particles/100 µL were isolated from AIS plasma with an average particle size of $158 \pm 10$ nm suggesting the presence of exosomes and/or microvesicles. The particle number did not vary considerably between healthy and AIS patients indicating EV NTA results alone are not sufficient for AIS diagnostics. The EV-mRNA assay provided 80% test positivity, similar to MRI (83%)[35] and vastly better than CT (26%). To improve the clinical sensitivity, larger gene panels can be used[7] along with enrichment of both CD8(+) and CD15(+) EVs; CD15 (+) neutrophils have been identified as a source of mRNA markers for AIS as well[7]. For the reported assay to be appropriate for clinical testing, the proper power analysis will need to be undertaken to secure clinical specificity and sensitivity to generate receiver-operator curves that also include hemorrhagic stroke patients and stroke mimics with the numbers determined from a power analysis.

While PEG precipitation of EVs provided a slightly higher mass yield of TRNA compared to EV-MAP (Fig. 4c), the challenge is that the EV-enriched fraction originates from all EV subpopulations and not only from the CD8+ fraction. As a consequence, mRNA expression profiles specific to the disease are obscured by mRNA from "non-diseased" EVs; Fig. 5e showed different gene expression profiles for sample #4 when using EVs enriched from the EV-MAP versus PEG. Only when EVs were secured using affinity isolation by EV-MAP was this sample correctly identified as an AIS patient.

We successfully developed a microfluidic with the ability to affinity-enrich and release EVs for NTA, TEM, and ddPCR analyses. The 7-bed EV-MAP, compared to the 3-bed device, increased the dynamic range for mRNA expression analysis and avoided bed saturation that may bias mRNA expression results. Owing to the beds' parallel configuration compared to the serial arrangement associated with the 3-bed device, input volumes of

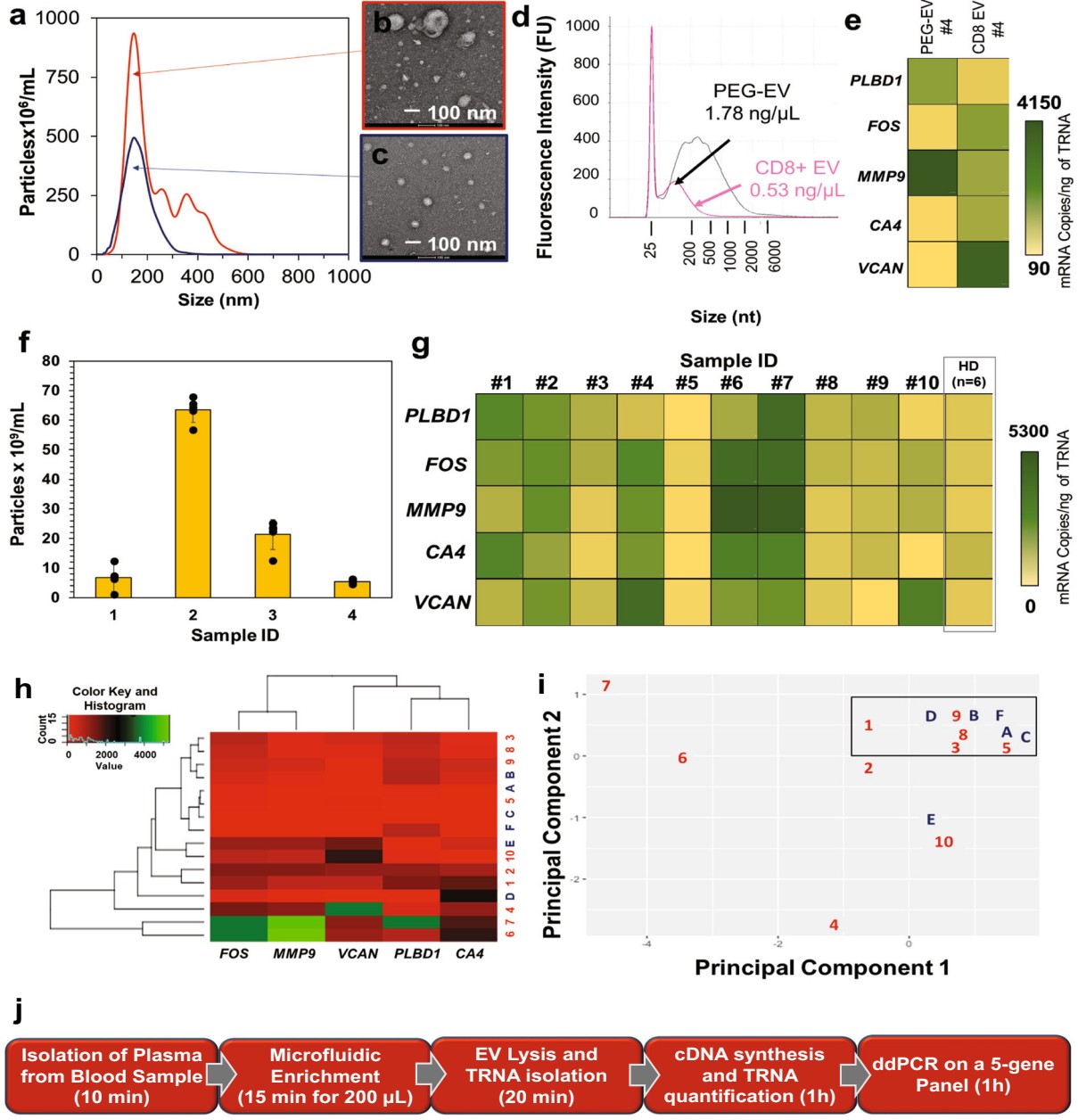

**Fig. 5 Affinity enrichment and mRNA transcripts analysis in CD8(+) EVs isolated from clinical samples. a** NTA and **b**, **c** TEM images of EVs isolated from clinical sample #4 by PEG precipitation and affinity selected with anti-CD8α mAb uisng the 7-bed EV-MAP. **d** Gel electrophoresis of TRNA size distributions from EVs isolated via EV-MAP and PEG precipitaion for sample #4. **e** Heat maps presenting the EV mRNA expression profiles for samlpe #4. **f** NTA results for selected samples 1, 4, 6, and 8. **g** mRNA expression profiling for selected genes in clinical samples. **h** Heat map analysis of clinical samples (marked with numbers) and healthy donors (identified with letters). **i** Principal component analysis for clinical samples (identified with numbers) and healthy donors (identified with letters). **j** Process flow chart showing the steps and time required for our EV mRNA expression profiling assay that uses the EV-MAP for EV enrichment and subsequent ddPCR quantification of five genes used for AIS diagnostics. The EV-MAP microfluidic for EV enrichment used the 7-bed device and accepted plasma samples with no pre-processing required (see Fig. 1d). Following enrichment, the EVs were released from the capture surface enzymatically (see Fig. 2), or directly lysed on the enrichment bed followed by solid-phase extraction (SPE) of the resulting total RNA (TRNA). mRNA was reverse transcribed and then subjected to ddPCR. Amounts of TRNA used in RT(+/−) are shown in Supplementary Table 7. The total processing time of this assay is 220 min (3.7 h), including the time for sampling and pipetting.

sample can be processed in a shorter time. For example, EV recovery of 70% was achieved at 20 μL/min volumetric flow using the 7-bed EV-MAP (Fig. 1g), while the 3-bed device's recovery at this volumetric flow rate was considerably poorer.

An advantage of using EVs compared to biological cells for mRNA profiling may be their higher abundance, supplying larger amounts of mRNA responding to an inflammatory event indicative of AIS in a shorter period of time. This has been shown in

early stage cancer diagnosis[36]. To verify the time-dependent evolution of mRNA profiles following AIS onset in EVs, an in vitro AIS animal model can be used[37]. In addition, while we have provided results from a pilot study, an AIS cohort with the appropriate power must be undertaken to validate the utility of using EV mRNA for AIS diagnosis. Such large clinical studies can be supported using the EV-MAP as it is fabricated in a high-scale production mode[38].

Several processing steps can be altered in the reported assay to reduce the processing time as well as increase the level of assay automation. The EV release process can be shortened using a coumarin-based photocleavable linker, enabling high efficiency release of EVs within 2 min without damaging the mRNA cargo[39]; 60 min was required for USER® enzyme cleavage of the uracil-containing oligonucleotide bifunctional linker[24]. The TRNA normalization strategy, which utilized gel electrophoresis (see Fig. 5d) for ddPCR, can be converted to a "per particle" normalization by utilizing an on-line chip-based nanopore counting strategy of EVs. We are also designing an integrated fluidic processor consisting of the EV-MAP, an in-plane nano-pore sensor, EV lysis with TRNA solid-phase extraction unit, a continuous flow reverse transcription reactor, and a nanosensor for the label-free quantification of specific mRNAs using a solid-phase ligase detection reaction. This will eliminate the need for specialized operators to carry out the molecular test through assay automation to enable mRNA expression profiling even at the point-of-care, which will provide more timely results to allow more AIS patients to receive rt-PA treatment.

It is the ultimate goal of our research work to design an assay and the associated hardware that can operate as a point-of-care test that can be performed by first responders and the results of which would be secured even before a potential patient reaches the hospital. Ideally, if the patient tested positive for AIS, he/she would be transported to a stroke center that offers the critical elements for more timely decisions on the administration of rt-PA therapy.

## Methods

**Materials and reagents**. Materials and reagents included COC cover plates (6013S-04) and substrates (5013L-10) (TOPAS Advanced Polymers), reagent-grade IPA (isopropyl alcohol), 1-ethyl-3-[3-dimethylamino-propyl] carbodiimide hydro-chloride (EDC), N-hydroxysuccinimide (NHS), 2-(4-morpholino)-ethane sulfonic acid (MES), BSA, Triton X-100, polyvinylpyrrolidone, 40 kDa (PVP-40), Histo-paque 1077, and PEG secured from Sigma-Aldrich. Phosphate buffered saline (PBS, pH = 7.4), RPMI-1640 medium, and FBS were purchased from Gibco laboratories. Sodium dodecyl sulfate (SDS), Micro-90, LIVE/DEAD Kit, and Toluidine Blue O were obtained from Fisher Scientific. TBS, Tween20, and EVA Green supermix were obtained from Bio-Rad. Other reagents included uranyl acetate (Polysciences), TapeStation supplies (Agilent), LPS from *Escherichia coli* 0111:B4 (LPS) (InvivoGen), Zymo RNA kit (Zymo Research), BCA Protein Assay Kit (Pierce), ProtoScript II First Strand cDNA Synthesis Kit (New England Bio-Labs). Antibodies used in these studies included anti-human CD8α mAbs (Clone # 37006), APC conjugated mouse anti-human CD8α mAb (Clone # 37006), FITC-labeled anti-CD45 mAb, and APC conjugated mouse IgG2B anti hCD8α Ab obtained from R&D systems/Biotechne.

**Cell line and growth conditions**. MOLT-3 T-cells (ATCC CRL-1552) were grown in RPMI-1640 medium supplemented with 10% FBS. The cells were grown at 37 °C in 5% CO$_2$. To remove the EVs from FBS, the serum was ultracentrifuged (LM8 Beckman Coulter Ultracentrifuge) at $100,000 \times g$ for 18 h. EV-depleted FBS was used in the cell culture media. Cell viability was monitored using a LIVE/DEAD kit.

**T-cell stimulus with LPS**. To model stroke conditions in cells, we stimulated MOLT-3 T-cells with LPS. The stock solution of LPS was prepared in sterile PBS. Then, the cells (culture started with approximately one million cells) were cultured with LPS and monitored for morphology changes and viability at time points up to 75 h. Control experiments were carried out without stimulating the cells with LPS. At each time point, cell viability was evaluated using the LIVE/DEAD viability/cytotoxicity kit for mammalian cells.

**Determination of protein concentration**. Protein concentration was evaluated with the BCA Protein Assay Kit (Pierce) according to the manufacturer's protocol. Calibration curves were constructed between 0.5 and 100 µg/mL BSA concentra-tion ($y = 0.0745x + 0.0936$, $R^2 = 0.998$). Protein samples were collected in the following way: RIPA Lysis buffer (1×) was infused into the EV-MAP at 10 µL/min and ~150 µL of lysate was collected. The solutions were immediately sampled by the BCA assay. Usually 5 µL of the sample was mixed with 95 µL of water and 100 µL of kit components (solution A, solution B, and solution C at a volumetric ratio of 25:24:1, respectively). The samples were incubated for 60 min at 60 °C. Following reaction, the absorption spectra for standards and samples were collected using a

UV–Vis spectrophotometer (Shimadzu) using absorption at a peak maximum of 560 nm.

**EV precipitation with PEG**. PEG precipitation of EVs was carried out as reported earlier[29]. In brief, the procedure was as follows: plasma sample was mixed in 0.5× volume of PBS and mixed with 2 mg/mL proteinase K. Proteins were digested for 20 min at 37 °C. An equal volume of PEG was added to the mixture of plasma, PBS, and proteinase K. After adding PEG, the tube was inverted and placed in 4 °C overnight before centrifuging the solution at $4000 \times g$ for 1 h at 4 °C. The pellet was lysed (vortexed thoroughly to dissolve the pellet completely) and RNA was isolated using the Zymo RNA kit according to the manufacturer's protocol.

**Microfluidic fabrication**. Three-bed EV-MAP devices were produced via hot embossing into COC from a molding master fabricated in brass via high-precision micromilling (HPMM). Details are provided in the Supporting Information. Seven-bed EV-MAP devices used in this study were fabricated in COP via injection molding (Stratec, Austria) from a mold insert made via UV-LiGA[40].

**Immobilization of mAbs**. Modification of COC and COP devices for affinity selection of cells and EVs employed a single-stranded oligonucleotide bifunctional cleavable linker containing a uracil residue that could be cleaved using a USER®[24]. Concentrations of anti-CD8α mAb were 1 mg/mL and 2 mg/mL for the 3-bed and 7-bed EV-MAP devices, respectively. Detailed procedures for mAb immobilization were reported previously[24].

**T-cells and EV affinity purification**. CD8(+) T-cells or CD8(+) MOLT-3 cells were isolated from healthy donor whole blood and cell media, respectively, using curvilinear channel devices modified with anti-human CD8α antibody[22]. The blood samples were collected into EDTA tubes to prevent the coagulation of blood and were analyzed on the same day the blood was collected. Two milliliters of MOLT-3 cells (~$1 \times 10^6$ cells/mL) was centrifuged for 10 min at $300 \times g$ to pellet the cells. The cell pellet was resuspended in 5 mL of PBS, and 1 mL of the sus-pension was infused onto a cell isolation chip. Both CD8(+) T-cells or CD8(+) MOLT-3 cells were isolated at 25 µL/min through the curvilinear channel device. Following cell affinity capture, the microfluidic device was washed with 1 mL of 0.5% BSA/PBS at 55 µL/min to remove unbound cells. For the enumeration, the cells were released from the device using USER® enzyme and collected in a glass bottom well of a 96-well plate. The cells were immunostained for identification and enumeration (see protocol in Supplementary Information).

EVs were isolated from plasma or cell media using either the 3- or 7-bed EV-MAP devices. To obtain plasma and medium appropriate for EV isolation, the blood components and cell suspensions were centrifuged at $300 \times g$ for 5 min followed by $1000 \times g$ for 10 min before the plasma or medium was infused into the microfluidic chip. All samples were hydrodynamically driven through the chip using a syringe pump (New Era Pump Systems, Inc., Farmingdale, NY, USA) and syringe fitted with a capillary connector. To minimize non-specific adsorption, mAb-modified EV-MAP surfaces were blocked with 1% polyvinylpyrrolidone (PVP) and 0.5% BSA in PBS (200 µL, 10 µL/min), then washed with 1% Tween20 in TBS after enrichment to remove non-specifically bound material. The cell media or plasma samples were infused into the 3-bed EV-MAP at 5 µL/min. Post-isolation rinse was performed at 10 µL/min with TBS/Tween20 (Bio-Rad, Hercules, CA). All the buffer solutions used for rinsing were filtered using a 0.45-µm polypropylene housing, surfactant-free cellulose acetate membrane filter (Thermo Scientific) prior to use.

**Fluorescence visualization of EVs membrane antigens**. Following EV isolation, devices were incubated with APC conjugated mouse anti-human CD8α mAb (R&D systems/BioTechne) for 40 min. As controls, the same procedure was carried out for an UV/O$_3$-modified device without anti-CD8α mAb, but following EV enrichment. Isotype control experiments were performed by incubating a micro-fluidic device with APC-conjugated mouse anti-CD8α mAb. The devices were washed with TBS/Tween20 to remove any excess dye-labeled mAb and washed with PBS prior to fluorescence imaging. The devices were visualized using a 200M inverted microscope (Zeiss) with a 20× objective (0.3 NA, Plan NeoFluar), XBO 75 Xe arc lamp, single band Cy5 filter set (Omega Optical), Cascade:1K EM-CCD camera (Photometric), and MAC 5000 stage (Ludl Electronic Products), all of which were computer-controlled via Micro-Manager. The final images were background subtracted and analyzed using Image-J software.

**Transmission electron microscopy**. Enriched and subsequently released EVs in ~150 µL of PBS/USER® were vortexed thoroughly and 5 µL of the EV samples were placed onto a grid carbon (Carbon Type-B, 300 mesh, Copper, TED PELLA, Inc., Redding, CA) film side for 20 min. Then, the grid was washed with deionized water. Next, the grid was placed for 10 s in 2% (w/v) uranyl acetate stain filtered with a 0.22-µm filter (Thermo Scientific, IL, USA), and blot dried. The grids were dried for at least 15 min before viewing through the microscope (FEI TECNAI F20 XT field emission transmission electron Microscope, 200 kV electron source - Schottky field emitter).

**Nanoparticle tracking analysis**. EVs enriched and subsequently released in ~150 µL of PBS/USER® from the microfluidic device were analyzed via NTA (Nanosight NT 2.3). The samples were diluted 100×, and just before analysis they were vortexed thoroughly. The instrument parameters used for the analysis were: Camera shutter 1206, camera gain 366, capture duration 90 s. Five videos were taken for each sample at a temperature of 25 °C. The flow cell was washed five times with PBS in between sample analysis. During the final wash with PBS, the video was monitored to check if there were any particles left in the flow cell. If particles were detected in the video, washing was continued until no particles were seen. NTA was also used to evaluate the EV release efficiency by performing two consecutive rounds of USER® enzyme release and quantification of particles following each round.

**Samples processing in stroke model experiments**. To assess the changes in mRNA transcript abundance following an inflammation event, the experiments were performed with LPS naïve and LPS-stimulated cells grown in FBS depleted of EVs. Cell cultures with ~$1 \times 10^6$ cells/mL were stimulated with LPS. After 24 h, the cell suspension was centrifuged for 10 min at $300 \times g$ to pellet the cells. The cell pellet was resuspended in 5 mL of PBS, and 1 mL of the suspension was infused at 25 µL/min through the curvilinear channel device. Following cell affinity capture, the microfluidic device was washed with 1 mL of 0.5% BSA/PBS at 55 µL/min to remove unbound cells. EVs processing was as follows: after centrifugation of the cell media, the supernatant was used as source of EVs. The same isolation protocol was used as in "T-cell and EV affinity purification" section.

Following affinity selection, TRNA was isolated from cells and EVs, reverse transcribed, and subjected to mRNA expression analysis. ddPCR provided absolute quantification of the target cDNA (i.e., mRNA).

TRNA extraction from cells and EVs followed the same protocol; lysis was performed with lysis buffer provided in Zymo RNA kit. The lysate was introduced into a purification column. Further steps were completed according to the manufacturer's protocol. Purified TRNA was eluted in ~8 µL of water. The profiles of extracted TRNA were analyzed and quantified using gel electrophoresis—an Agilent 2200 TapeStation using 2 µL of eluent.

**cDNA synthesis from purified RNA**. Purified RNA isolated from cells or EVs was eluted from the purification column in ~8 µL of water. Two microliters of the eluent was taken for RNA quantification, and the remaining solution was used for the complementary DNA (cDNA) synthesis (2 µL for each RT(+) and RT(−) reactions in 20 µL total volume). cDNA was synthesized via reverse transcription (RT) reaction with poly-dT primer using ProtoScript II First Strand cDNA Synthesis Kit according to the manufacturer's instructions. Negative RT control reactions were performed in the absence of the enzyme.

**Droplet digital PCR**. Synthesized cDNA was used in ddPCR for gene expression analysis. The procedure for the PCR reaction preparation with 2 µL of the cDNA in 20 µL total reaction volume and 0.125 µM concentration of the primers followed manufacturer's suggestions. EVA Green Supermix (BioRad) was used for the PCR mix preparation. The primers for the genes vFOS (FBJ murine osteosarcoma viral oncogene), VCAN (Versican), PLBD1 (phospholipase B domain containing 1), MMP9 (metallopeptidase 9), and CA4 (carbonic anhydrase 4) were purchased from Integrated DNA Technologies. The primer sequences are given Supplementary Table 4. Droplet formation of the PCR mix in oil was performed with QX200 Droplet Generator according to manufacturer's protocol. PCR reactions were carried out in a C1000 touch thermal cycler (BioRad) with the following steps: 95 °C for 5 min; 40 cycles of denaturation at 95 °C for 30 s; annealing at 52 °C for 30 s; and extension at 72 °C for 1 min. A final cooling step was carried out at 4 °C. To read the droplets, a BioRad QX-200 ddPCR system was used, with the data analyzed using the QuantaSoft™ software. ddPCR results were normalized to ng of TRNA.

**Clinical samples**. Blood samples from healthy donors and plasma samples for the AIS patients were obtained from the Biorepository at the University of Kansas Medical Center, Kansas City, KS, and SUNY Down State Medical Center, New York, NY, respectively. Dr. Alison Baird lab personnel at SUNY Downstate Medical Center collected the clinical plasma samples according to this institution's IRB protocol. Informed consent was obtained from all patients. Plasma samples were stored at −80 °C until analysis.

**Analysis of clinical samples**. Ten single-blinded clinical samples, out of which five were AIS patient samples and five were healthy controls, were analyzed. The devices were pretreated in the same way as the 3-bed device. Following EV enrichment and wash, the selected EVs were either lysed on chip and TRNA extracted or released for NTA and TEM analysis (see protocols listed above). The yields of the TRNA isolated from clinical samples are presented in Supplementary Table 7.

**Statistics and reproducibility**. Statistical analysis using R Studio software was performed to identify patient samples. Heat maps were generated and PCA was

done for the ten clinical samples and additional datasets from six healthy donor plasma.

## Data availability

All data generated or analyzed during this study are included in this published article (and its Supplementary information files).

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

## Acknowledgements

The authors thank the NIH for financial support of this work via grants (P41-EB020594; P20-GM130423; R43-CA232848; R01-EB010087). The authors also acknowledge the KU Nanofabrication Facility (P20-GM103638), the University of Kansas Cancer Center's Biospecimen Repository Core (P30-CA168524), and the Microscopy and Analytical Imaging Laboratory at KU. We express gratitude to Dr. Davin Koestler for suggestions on statistical analysis of the data. We are very grateful to Mr. Nishantha Wijesuriya for his help in securing NTA data. C. Kramer acknowledges financial support from the Emerging Scholars Program at KU. We are very appreciative of patients and healthy donors for their blood donations.

## Author contributions

H.W. fabricated, assembled, and modified the devices, performed the affinity isolation experiments, NTA and TEM measurements, gene expression profiling, analyzed the data, and wrote the manuscript. M.A.W. designed the experiments, co-designed the 3-bed EV-MAP, performed the affinity isolation experiments, interpreted the data, and wrote the manuscript. J.M.J. modeled the EV affinity enrichment, designed the 3-bed EV-MAP, performed the NTA analyses, and wrote the manuscript. M.L.H. designed, assembled, and performed the evaluation of the 7-bed EV-MAP. V.B. and A.E.D. fabricated the 3-bed EV-MAP devices, K.H. modeled the EV affinity enrichment, C.K. performed the BCA assay. Y.L. secured and deidentified the AIS samples, A.E.B. secured the IRB, patient plasma samples, interpreted the data, and assisted in writing the manuscript. M.C.M. assisted in the design of the microfluidic device for EV isolation. S.A.S. assisted in the design and fabrication of the microfluidic for EV isolation, designed the experiments, interpreted the data, and assisted in writing the manuscript. All authors read the manuscript and approved of its content.

## Competing interests

S.A.S. has equity in BioFluidica, Inc., serves as the CSO and is on the Board of Directors for BioFluidica, Inc. M.L.H. is an employee of BioFluidica, Inc. and has an equity in the company. M.A.W. is a spouse of the employee of BioFluidica, Inc. Other authors declare no competing financial or non-financial interests.
