## [Peer Review File · Communications Biology]

Reviewers' comments:

Reviewer #1 (Remarks to the Author):

This study presents a number of important steps to detect a minority fraction of extracellular vesicles to a particular disease state. In order to enrich a minority fraction new microfluidic devices were designed, prepared and implemented. The microfluidic devices were experimentally tested and their effectivity to trap target EVs was simulated. The efficiency of the device was compared with other methods to non-specifically collect EVs. The EVs were trapped on the walls of the microfluidic device through a multifunctional spacer, that simultaneously enabled high density surface attachment, EV trapping/binding and release of EVs through enzymatic cleavage. The pool of enriched EVs was available for imaging methods (TEM) and nanoparticle tracking analysis (NTA) to investigate the size distribution of selected EVs. Furthermore, the pool of EVs was available for RNA analysis. The RNA content was analyzed for the presence of 5 types of mRNA previously associated with the onset of acute ischemic stroke (AIS). Every step in the analytical process is supported with convincing data and well-chosen control experiments. The paper is well written and the approach and devices proposed by the authors seems to be transferrable to other diagnostic challenges, such as those related to cancer detection and diagnostics. Especially the daunting fight to extract minority populations of EVs from the abundance of EVs may be greatly facilitated by microfluidics devices as proposed here by the authors.

The authors also mention, in several places of the manuscript, the ability of the method to limit the time for diagnosis to a critical time window within which medication against AIS should be started. This therefore certainly seems to be an asset, although maybe less relevant for, for instance, cancer diagnosis. I have no further comments.

Reviewer #2 (Remarks to the Author):

Wijerathne et al describe an in vitro assay based on the quantification of CD8(+) EVs and subsequent mRNA expression profiling of their mRNA cargo. The use of mRNA as opposed to proteins is advantageous because mRNA transcription precedes protein translation and allows for a more rapid detection in peripheral blood. The authors propose that this assay is rapid enough (3.7 h) to be used in a clinical setting to identify patients with ischemic stroke who would benefit from thrombolytic treatment, where the window to treatment is 4.5 h.

The assay seems robust and well-designed, and the use of mRNA rather than proteins is intriguing due to the time aspect in ischemic stroke treatment. However, I am ambiguous to the use of this assay in a clinical setting. The validation in ischemic stroke patients vs. healthy in this study is also insufficient and does not allow any conclusions.

Minor comments

- The results section is very long. Large parts of this section could be moved to methods.
- Ref 7 cannot be found
- The references to early mRNA expression (page 2 and 10) should read <3 h and not > 3h.

Major comments

- The stroke model described (Pg 7) is not a stroke model, but rather a model of inflammation.
- The sample size used to validate this assay in ischemic stroke patients vs healthy is far too small. The control group should also comprise patients with hemorrhage stroke or patients admitted due to stroke symptoms (stroke mimics) rather than healthy controls, as this is where the assay is proposed to be used.

- The time-line presented in Fig 5g is not realistic. According to the method section (supplements), samples were first spun 10 min twice (first box in figure). Even if isolation of plasma would be performed in a 10 min spin (as depicted in fig), sampling, pipetting etc needs to be taken into account.
- An assay which can be performed in 3.7 h would not be clinically helpful in the setting of thrombolytic treatment in acute ischemic stroke patients. A minority of patients reach the hospital within 30 min from symptom onset, and even in this minority, the delay for the test results is not justified. Although this assay may be diagnostic of ischemic stroke, the authors need to address another clinical setting.

Reviewer #3 (Remarks to the Author):

The manuscript describes an assay intended to provide rapid identification of acute ischemic stroke by microfluidic enrichment of CD8-positive extracellular vesicles, followed by identification of specific mRNA sequences that inform upon gene expression of T-cells. The key novel technology featured in the paper is a molded polymer micropillar filter device, in which capture antibodies are immobilized upon the enhanced surface area, resulting in efficient capture of specific EVs that display the matching protein on their outer surface. The investigators have engineered the device to provide a greater degree of sample throughput than previous implementations.

The method is intended to either complement or replace CT or MRI medical imaging of stroke patients, which have limitations in cost, sensitivity, and availability, and there is not presently an in vitro diagnostic test for AIS. The working hypothesis is that CD8+ T-cells will rapidly respond to the AIS through gene expression, and that upregulation of certain mRNA will provide a more rapid response than protein expression in peripheral blood. Further, the authors hypothesize that EVs produced by the AIS-responsive subpopulation of CD+ T-cells will carry a representative mRNA cargo, and that they can be gathered from peripheral blood for analysis.

Because I am not a biologist, I unfortunately cannot address the validity of the above hypothesis, however it does raise a concern that the authors may want to clarify for the readership: If a stroke occurs at $t=0$, and a blood sample must be obtained as close to $t=0$ as possible to have ample time for performing the assay, thus under the most optimal scenario a blood sample can be gathered at $t=30$ min. At that point, has enough time elapsed for CD8+ T-cells to respond to the AIS event, and can sufficient EVs be produced for detection? Assuming that CD8+ T-cells continuously shed EVs, what fraction of all the CD8+ EVs could be those that have altered mRNA content? Is it possible that the post-AIS gene expression from CD8+ EVs represents a more chronic condition that existed before the acute event? If so, is that information still relevant/important for correctly identifying the cause of the stroke and providing therapy?

That said, the manuscript clearly describes the design of the EV-MAP device, its preparation for capturing CD8+ EVs with a high degree of efficiency, the ability to release/gather the captured EVs, and for the steps to be performed rapidly using reasonable flow rates. The recovery is facilitated by a clever oligo bifunctional linker that can be chemically triggered to release its attachment from the EV-MAP surface.

The authors demonstrate operation of the device initially with a cell line model for stroke, in which an inflammatory response can be induced, and the resulting EVs extracted from the cell media. A set of genes are profiled by ddPCR that show upregulation in response to the stimulus. Gene expression is compared between T-cells and EVs from 6 healthy patients. Using a small sample population of AIS patients, gene expression profiles from purified EVs could be differentiated from healthy controls using principle component analysis. The results look encouraging although a larger sample set would be needed to draw stronger conclusions.

Overall, the manuscript is well-written, the experimental protocols were clearly described, and the figures are excellent. I recommend acceptance of the manuscript, and I recommend that the authors incorporate some discussion regarding the third paragraph of my review. I observed one typographical error on page 3, line 88: "selecy"

Manuscript ID: COMMSBIO-20-0348-T
Responses to the Reviewers

We sincerely thank the reviewers for their careful evaluation of our manuscript and the helpful comments provided to improve the quality of this work. We have modified the manuscript according to requests by the editor and reviewers. The clinical application of our assay is presented in more subtle terms in the revised version of this manuscript. We have also modified the title to better reflect the presented work and deemphasize clinical references. Below, please find our responses to questions/comments provided by the reviewers. All changes made in the revised manuscript are highlighted in yellow.

Reviewer #1:

This study presents a number of important steps to detect a minority fraction of extracellular vesicles to a particular disease state. In order to enrich a minority fraction new microfluidic devices were designed, prepared and implemented. The microfluidic devices were experimentally tested and their effectivity to trap target EVs was simulated. The efficiency of the device was compared with other methods to non-specifically collect EVs. The EVs were trapped on the walls of the microfluidic device through a multifunctional spacer, that simultaneously enabled high density surface attachment, EV trapping/binding and release of EVs through enzymatic cleavage. The pool of enriched EVs was available for imaging methods (TEM) and nanoparticle tracking analysis (NTA) to investigate the size distribution of selected EVs.

Furthermore, the pool of EVs was available for RNA analysis. The RNA content was analyzed for the presence of 5 types of mRNA previously associated with the onset of acute ischemic stroke (AIS). Every step in the analytical process is supported with convincing data and well-chosen control experiments. The paper is well written and the approach and devices proposed by the authors seems to be transferrable to other diagnostic challenges, such as those related to cancer detection and diagnostics. Especially the daunting fight to extract minority populations of EVs from the abundance of EVs may be greatly facilitated by microfluidics devices as proposed here by the authors. The authors also mention, in several places of the manuscript, the ability of the method to limit the time for diagnosis to a critical time window within which medication against AIS should be started. This therefore certainly seems to be an asset, although maybe less relevant for, for instance, cancer diagnosis. I have no further comments.

Answer: We appreciate the positive comments on our work. No direct questions or comments were provided by this reviewer.

Reviewer #2:

Wijerathne et al describe an in vitro assay based on the quantification of CD8(+) EVs and subsequent mRNA expression profiling of their mRNA cargo. The use of mRNA as opposed to proteins is advantageous because mRNA transcription precedes protein translation and allows for a more rapid detection in peripheral blood. The authors propose that this assay is rapid enough (3.7 h) to be used in a clinical setting to identify patients with ischemic stroke who would benefit from thrombolytic treatment, where the window to treatment is 4.5 h.

The assay seems robust and well-designed, and the use of mRNA rather than proteins is intriguing due to the time aspect in ischemic stroke treatment.

However, I am ambiguous to the use of this assay in a clinical setting. The validation in ischemic stroke patients vs. healthy in this study is also insufficient and does not allow any conclusions.

Answer: Thank you for the critique. We have modified the conclusions regarding the clinical

application of our assay. We state that further clinical testing must be undertaken using a powered analysis to assess the utility of the method in a clinical setting. Below highlights changes we have made to the revised manuscript to express this view point.

Page 11: “For the reported assay to be appropriate for clinical testing, the proper power analysis will need to be undertaken to secure clinical specificity and sensitivity to generate receiver-operator curves that also include hemorrhagic stroke patients and stroke mimics with the numbers determined from a power analysis.”

Page 12: “To verify the time-dependent evolution of mRNA profiles following AIS onset in EVs, an *in vitro* AIS animal model can be used.³⁷ In addition, while we have provided results from a pilot study, an AIS cohort with the appropriate power must be undertaken to validate the utility of using EV mRNA for AIS diagnosis.”

Minor comments

- The results section is very long. Large parts of this section could be moved to methods.

Answer: We edited the text and moved sections into the methods and SI of the document. Moved sections are highlighted.

- Ref 7 cannot be found

Answer: We included the following link associated with reference 7: <https://journals.ke-i.org/mra/article/view/1597/1583>

- The references to early mRNA expression (page 2 and 10) should read <3 h and not > 3h.

Answer: We corrected errors in the document. Thank you for noticing this.

Major comments

- The stroke model described (Pg 7) is not a stroke model, but rather a model of inflammation.

Answer: We made appropriate changes and described this work as “inflammation model.” We clarified the wording in the manuscript.

Page 7: “EV mRNA expression in an inflammation model. By evaluating the response of MOLT-3 cells and the EVs they generate when exposed to lipopolysaccharide (LPS), we mimicked the inflammation process during an AIS event.”

- The sample size used to validate this assay in ischemic stroke patients vs healthy is far too small. The control group should also comprise patients with hemorrhage stroke or patients admitted due to stroke symptoms (stroke mimics) rather than healthy controls, as this is where the assay is proposed to be used.

Answer: This study was performed as a pilot to investigate the applicability of the EV-mRNA changes for detection of ischemic stroke. In follow up clinical studies, we will expand the work to accommodate a larger sample size, and we will include the necessary controls consisting of not only a healthy group but also hemorrhagic stroke patients and stroke mimics. We added this wording in the discussion section.

Please, see page 11 and 12 of the edited manuscript as noted above.

- The time-line presented in Fig 5g is not realistic. According to the method section

(supplements), samples were first spun 10 min twice (first box in figure). Even if isolation of plasma would be performed in a 10 min spin (as depicted in fig), sampling, pipetting etc needs to be taken into account.

Answer: *Time for sampling and pipetting was included in calculating the total time for this assay. We clarified this in the caption of Figure 5g.*

- An assay which can be performed in 3.7 h would not be clinically helpful in the setting of thrombolytic treatment in acute ischemic stroke patients. A minority of patients reach the hospital within 30 min from symptom onset, and even in this minority, the delay for the test results is not justified. Although this assay may be diagnostic of ischemic stroke, the authors need to address another clinical setting.

Answer: *It is the ultimate goal of our research work to design an assay and the associated hardware that can operate as a point-of-care test that can be performed by first responders and the results of which would be secured even before a potential patient reaches the hospital. Ideally, if the patient tested positive for AIS, he/she would be transported to a stroke center that offers the critical elements for more timely decisions on the administration of rt-PA therapy.*

This information was added to the revised manuscript on page 13. We should note as well that in some cases, AIS can be treated with endovascular thrombectomy, which has an effective therapeutic window of ~7 h; we now have introduced this into the revised manuscript and have included the appropriate references.

We include this section in the manuscript – please see page 12.

Reviewer #3:

The manuscript describes an assay intended to provide rapid identification of acute ischemic stroke by microfluidic enrichment of CD8-positive extracellular vesicles, followed by identification of specific mRNA sequences that inform upon gene expression of T-cells. The key novel technology featured in the paper is a molded polymer micropillar filter device, in which capture antibodies are immobilized upon the enhanced surface area, resulting in efficient capture of specific EVs that display the matching protein on their outer surface. The investigators have engineered the device to provide a greater degree of sample throughput than previous implementations.

The method is intended to either complement or replace CT or MRI medical imaging of stroke patients, which have limitations in cost, sensitivity, and availability, and there is not presently an in vitro diagnostic test for AIS. The working hypothesis is that CD8+ T-cells will rapidly respond to the AIS through gene expression, and that upregulation of certain mRNA will provide a more rapid response than protein expression in peripheral blood. Further, the authors hypothesize that EVs produced by the AIS-responsive subpopulation of CD+ T-cells will carry a representative mRNA cargo, and that they can be gathered from peripheral blood for analysis.

Because I am not a biologist, I unfortunately cannot address the validity of the above hypothesis, however it does raise a concern that the authors may want to clarify for the readership: If a stroke occurs at $t=0$, and a blood sample must be obtained as close to $t=0$ as possible to have ample time for performing the assay, thus under the most optimal scenario a blood sample can be gathered at $t=30\text{min}$. At that point, has enough time elapsed for CD8+ T-cells to respond to the AIS event, and can sufficient EVs be produced for detection?

Answer: Thank you for this comment. Future studies will answer this question. Work with animal models may allow us to collect blood samples at different time points following the stroke event. Analysis of the EV-mRNA at each time point will be used to determine the time at which we identify release of “activated” CD8+ EVs. We elaborated on it in the manuscript.

Page 12: To verify the time-dependent evolution of mRNA profiles following AIS onset in EVs, an *in vitro* animal model can be used to determine the kinetics of marker appearance following AIS.

Assuming that CD8+ T-cells continuously shed EVs, what fraction of all the CD8+ EVs could be those that have altered mRNA content?

Answer: We appreciate this question. We suppose the abundance of CD8+ EVs would be dependent on the fraction of activated CD8+ T-cells following the AIS event and the time following the AIS. A better understanding of the process of CD8+ T-cell rate of release of EV following the AIS would indeed be important to determine in our future work. To answer this question, however, carefully designed animal model studies should be performed, so the kinetics of the CD8+ T-cells activation and post-activation release of EV with altered mRNA cargo can be determined.

We emphasized this in the main text. Please see page 10.

Page 10: “We have shown that expression of certain genes specific to AIS in CD8(+) T-cells provided 66%, 87%, and 100% clinical sensitivity for 2.4 h, 5 h, and 24 h following stroke onset, suggesting that mRNA expression differences indicative of AIS have improved clinical sensitivity with time as more leukocytes respond to an inflammatory insult.”

Is it possible that the post-AIS gene expression from CD8+ EVs represents a more chronic condition that existed before the acute event? If so, is that information still relevant/important for correctly identifying the cause of the stroke and providing therapy?

Answer: The following response was added to the revised manuscript (page 3): “The contribution of PBMCs’ mRNA expression changes in both pre-existing and an acute response to stroke has been studied by Moore *et al.*, indicating a partial dependence of expression changes on pre-existing vascular risk conditions. Hence, there may be a contribution of both risk and response to acute stroke on the mRNA expression profile. However, this would not diminish the value of the information concerning the acute event.”

That said, the manuscript clearly describes the design of the EV-MAP device, its preparation for capturing CD8+ EVs with a high degree of efficiency, the ability to release/gather the captured EVs, and for the steps to be performed rapidly using reasonable flow rates. The recovery is facilitated by a clever oligo bifunctional linker that can be chemically triggered to release its attachment from the EV-MAP surface.

The authors demonstrate operation of the device initially with a cell line model for stroke, in which an inflammatory response can be induced, and the resulting EVs extracted from the cell media. A set of genes are profiled by ddPCR that show upregulation in response to the stimulus. Gene expression is compared between T-cells and EVs from 6 healthy patients. Using a small sample population of AIS patients, gene expression profiles from purified EVs could be differentiated from healthy controls using principle component analysis. The results look encouraging although a larger sample set would be needed to draw stronger conclusions.

Overall, the manuscript is well-written, the experimental protocols were clearly described, and the figures are excellent. I recommend acceptance of the manuscript, and I recommend

that the authors incorporate some discussion regarding the third paragraph of my review. I observed one typographical error on page 3, line 88: "selecy"

Answer: *We corrected the error. Thank you.*

REVIEWERS' COMMENTS:

Reviewer #2 (Remarks to the Author):

-

Reviewer #3 (Remarks to the Author):

The authors kindly addressed the issues I raised in my review. I recommend acceptance of the paper.